# Renal Proximal Tubule Cell Cannabinoid-1 Receptor Regulates Bone Remodeling and Mass via a Kidney-to-Bone Axis

**DOI:** 10.3390/cells10020414

**Published:** 2021-02-17

**Authors:** Saja Baraghithy, Yael Soae, Dekel Assaf, Liad Hinden, Shiran Udi, Adi Drori, Yankel Gabet, Joseph Tam

**Affiliations:** 1Obesity and Metabolism Laboratory, The Institute for Drug Research, School of Pharmacy, Faculty of Medicine, The Hebrew University of Jerusalem, Jerusalem 9112001, Israel; saja.baraghithy@mail.huji.ac.il (S.B.); yael.soae@gmail.com (Y.S.); dekel.assaf@mail.huji.ac.il (D.A.); liad77@gmail.com (L.H.); shiran.udi@mail.huji.ac.il (S.U.); adidrori10@gmail.com (A.D.); 2Department of Anatomy and Anthropology, Sackler Faculty of Medicine, Tel-Aviv University, Tel-Aviv 69978, Israel; yankel@tauex.tau.ac.il

**Keywords:** type 1 diabetes, osteoporosis, CB1 receptor, erythropoietin

## Abstract

The renal proximal tubule cells (RPTCs), well-known for maintaining glucose and mineral homeostasis, play a critical role in the regulation of kidney function and bone remodeling. Deterioration in RPTC function may therefore lead to the development of diabetic kidney disease (DKD) and osteoporosis. Previously, we have shown that the cannabinoid-1 receptor (CB_1_R) modulates both kidney function as well as bone remodeling and mass via its direct role in RPTCs and bone cells, respectively. Here we employed genetic and pharmacological approaches that target CB_1_R, and found that its specific nullification in RPTCs preserves bone mass and remodeling both under normo- and hyper-glycemic conditions, and that its chronic blockade prevents the development of diabetes-induced bone loss. These protective effects of negatively targeting CB_1_R specifically in RPTCs were associated with its ability to modulate erythropoietin (EPO) synthesis, a hormone known to affect bone mass and remodeling. Our findings highlight a novel molecular mechanism by which CB_1_R in RPTCs remotely regulates skeletal homeostasis via a kidney-to-bone axis that involves EPO.

## 1. Introduction

The skeleton is not merely a solid protective frame that facilitates locomotion—it is also an intricate endocrine system that regulates numerous essential physiological functions such as energy and mineral metabolism [1,2]. Indeed, mineral homeostasis is regulated via a crosstalk across the bone–kidney–intestine axis to ensure that the intra- and extracellular levels of calcium and phosphate remain within a relatively narrow range, via an interplay of three major hormones, i.e., parathyroid hormone (PTH), the active metabolite of vitamin D, 1,25 dihydroxyvitamin D, calcitonin, and fibroblast growth factor 23 (FGF-23). These hormones also regulate the serum levels of each other; thus, dysregulation of one of them through this axis may cause the dysregulation of all and consequently, induce bone loss [3,4].

Recent evidence shows that the hormonal regulation of bone mineral homeostasis is, however, much more complex and involves several systemic, kidney-specific, or bone-specific factors, such as sex hormones, insulin, cortisol, glucocorticoids, calcitonin, bone morphogenic proteins (BMPs), Klotho, erythropoietin (EPO), and numerous cytokines [5,6,7,8]. A plethora of disease states such as chronic kidney disease (CKD), diabetes, obesity, genetic disorders, and aging may cause the dysregulation of these factors and, consequently, may induce adverse effects on mineral metabolism and bone health. Thus, a thorough understanding of specific pathways affecting kidney-to-bone relationships is essential for targeted and innovative therapeutics.

Type 1 diabetes (T1D) is associated with major microvascular (retinopathy, nephropathy, and neuropathy) and macrovascular (atherosclerosis) complications. Additionally, T1D is associated with a low bone mineral density (BMD) and an increased risk of fracture, resulting from ablation of the insulin bone-anabolic effect, the toxic hyperglycemia, inflammation, oxidative stress, and superfluous urinary calcium excretion [9,10,11]. Furthermore, bone loss associated with T1D can be secondary to diabetic kidney disease (DKD), which affects approximately 30% of subjects with T1D and can eventually progress to end-stage renal disease (ESRD) [12]. DKD is characterized by a decreased glomerular filtration rate (GFR), glomerular and tubular hypertrophy, albuminuria, renal inflammation, and tubulointerstitial fibrosis. This gradual deterioration in renal function is subsequently accompanied by bone impairment due to hypocalcemia, deficiency of vitamin D, as well as secondary hyperparathyroidism [13].

For the past decade our research group has focused on elucidating the role of the endocannabinoid (eCB) system in several physiological/pathological functions. Our findings indicate that this endogenous system has crucial effects on the maintenance of both glucose and bone homeostasis; thus, deviation from this homeostasis can cause two major disorders: Diabetes and osteoporosis, suggesting that modulating the activity of the eCB system holds therapeutic promise for treating these disorders [14,15,16,17]. Moreover, the eCB system is also involved in regulating kidney function. Specifically, the kidneys are a major source of eCBs, and the main eCB receptor (CB_1_R) is vastly expressed in many cell types in the kidney [18]. Among them, the renal proximal tubular cells (RPTCs) are critically important in regulating normal kidney function, since they are responsible for 80% of the renal reabsorption, and are particularly sensitive to the deleterious effect of chronic hyperglycemia in diabetic patients and animals because glucose enters these cells independently of insulin. In diabetic and obese conditions there is a marked increase in the eCB/CB_1_R “tone”, which we recently found to promote kidney dysfunction, injury, inflammation, oxidative stress, mitochondrial fragmentation, and fibrosis by compromising the function of the RPTCs [18,19,20,21,22]. Moreover, pharmacological blockade of CB_1_R was found to ameliorate these kidney abnormalities, thus preventing the development of DKD [18,20].

Considering all these findings, we set out to examine the contribution of renal CB_1_R in the etiology of bone loss resulting from T1D. Here, we describe a novel pathway by which CB_1_R, specifically in RPTCs, remotely regulates bone mass and function under physiological and pathological conditions via the kidney-to-bone mediator EPO. The overall findings of this study may indicate that targeting renal CB_1_R activity and/or EPO may provide a promising therapeutic option for the treatment of T1D-induced bone loss.

## 2. Materials and Methods

### 2.1. Materials

Streptozotocin (STZ), the CB_1_R agonists arachidonyl-2′-chloroethylamide (ACEA), noladin ether (NE), and anandamide (AEA), as well as the fatty acid amide hydrolase (FAAH)/monoacylglycerol lipase (MAGL) inhibitor JZL-195 were purchased from Cayman Chemicals. The CB_1_R antagonists SLV-319 and JD5037 were purchased from Haoyuan Chemexpress Co., Ltd (Shanghai, China). Pertussis toxin (PTX) was purchased from Sigma–Aldrich/Merck (Jerusalem, Israel).

### 2.2. Animals

All experimental protocols used here were approved by the Institutional Animal Care and Use Committee of the Hebrew University of Jerusalem (AAALAC accreditation #1285; ethic approval numbers MD-14-14121-4, MD-15-14198-3, and MD-16-14683-4). Animal studies are in compliance with the ARRIVE guidelines [23], and are based on the rule of the replacement, refinement, or reduction. All the animals used in this study were housed under specific pathogen-free (SPF) conditions, up to five per cage, in standard plastic cages with natural soft sawdust as bedding. The animals were maintained under a controlled temperature of 22–24 °C, humidity at 55 ± 5%, and alternating 12-h light/dark cycles (lights were on between 7:00 and 19:00 h); animals were provided with a standard diet (14% fat, 24% protein, 62% carbohydrates; NIH-31 rodent diet) and water ad libitum. All of the studies were performed using male mice on the C57Bl/6J background. RPTC-CB_1_^−/−^ mice were generated by crossing mice containing two loxP sites flanking the open reading frame of the CB_1_R with the iL1-sglt2-Cre line, which is specifically expressed in the RPTCs, as described previously [22]. Mice were genotyped by PCR. For the iL1-sglt2-Cre line, the primers 5′-CAG GGT GTT ATA AGC AAT CCC-3′ and 5′-CCT GGA AAA TGC TTC TGT CCG-3′ produced a band of 350 bp, when the transgene was present. For the genotyping of the *Cnr1* gene locus, G50, G51, and G53 primers were used as described previously [24].

### 2.3. Experimental Procedure

In studies requiring the induction of T1D, male 8-week-old C57Bl/6J or RPTC-CB_1_^−/−^ mice and their littermate controls were administered five consecutive intraperitoneal (IP) injections of STZ (50 mg/kg per day; Cayman Chemical). Two weeks after the first injection, blood glucose levels were measured using a glucometer (Contour Bayer, Germany), and animals with blood glucose levels greater than 250 mg/dL were considered diabetic and therefore were included in the study. Control groups for each strain were given 0.1 mol/L citrate buffer (pH 4.5). STZ-induced diabetic C57Bl/6J mice were treated daily with a global CB_1_R antagonist SLV-319 (3 mg/kg, orally), or vehicle (Veh; 1% Tween80 and 4% DMSO in saline) for 16 weeks. Body weights and glucose levels were monitored weekly. To assess the in vivo effects on bone mineralization, newly mineralized bone was vitally labeled by fluorochrome calcein (Sigma–Aldrich/Merck, Jerusalem, Israel) that was injected IP at 15 mg/kg 4 and 1 d before euthanasia. At the age of 26 weeks, the mice were euthanized by a cervical dislocation under anesthesia. Once euthanized, trunk blood was collected for determining the biochemical parameters, kidneys were taken and kept at −80 °C, the femoral bones and L3 vertebrae were separated, cleaned, and fixed in 10% phosphate buffered formalin (pH 7.2) for 48 h, and then kept in 70% ethanol until further use.

### 2.4. Biochemistry

Total calcium (CA2), alkaline phosphatase (ALP2S), and phosphate (PHOS) were determined in serum or urine (as defined in the figures) using the Cobas C-111 chemistry analyzer (Roche, Switzerland). Serum osteocalcin levels were determined using an EIA kit (60-1305; Immutopics, Inc.). Mouse C-telopeptide of type I collagen (CTX-1) was measured in the same specimens using an EIA kit (AC-06F1; Immunodiagnostic Systems). Serum insulin levels were determined by ELISA (EZRMI-13K; Merck Millipore).

### 2.5. Skeletal Phenotyping

The skeletal phenotype was analyzed by a combinatorial µCT/histomorphometric system as described previously [25]. Briefly, femora and L3 lumbar vertebrae were examined using a μCT system (μCT 40; Scanco Medical AG) at a 10-μm isotropic resolution, with an X-ray tube potential of 70 kVp, intensity of 114 µA, and integration time of 200 ms. In the femora, trabecular bone parameters were measured in the distal metaphyseal segment, extending 3 mm proximally from the proximal tip of the primary spongiosa. Cortical bone parameters were determined in a diaphyseal segment extending 1 mm distally from the midpoint between the femoral ends. Trabecular bone parameters were also analyzed in L3 bodies. After μCT image acquisition, the femoral specimens were embedded undecalcified in polymethylmethacrylate (Technovit 9100; Heraeus Kulzer, Wehrheim, Germany). Longitudinal sections through the midfrontal plane were left unstained for dynamic histomorphometric analyses, based on the vital calcein double labeling. To identify osteoclasts, consecutive sections were deplasticized and stained for tartrate-resistant acid phosphatase (TRAP; Sigma–Aldrich/Merck, Jerusalem, Israel), and counterstained with Mayer’s hematoxylin. Histomorphometric analysis was carried out on digital photomicrographic images using the IMAGE-PRO PLUS V.6 software (Media Cybernetics, Rockville, MD, USA). The following parameters were determined: Mineral apposition rate (MAR), mineralizing surface per bone surface (MS/BS), bone formation rate (BFR/BS), and osteoclast number (N.Oc/BS). The terminology used for these measurements was according to the convention of standardized nomenclature [26].

### 2.6. Mechanical Testing

A three-point bending test was performed as reported previously [27]. Briefly, femora were re-hydrated in PBS to restore the mechanical properties of the tissue 24 h before biomechanical testing. The three-point bending test was performed using a loading machine (model 4502; Instron) equipped with a 100 N load cell at a cross-head speed of 1 mm/min. The bone specimens were placed in a custom-made device with a span length of 20 mm and loaded until failure, and force vs. deflection data were acquired automatically. The load-deflection curve yielded three parameters for evaluation: Total energy (Nmm) applied on the bone up to ultimate/fracture load, calculated by the AUC; ultimate load (N); and bending stiffness (N/mm), calculated as the slope of the load-deflection curve at its linear portion.

### 2.7. Cell Culture

Human immortalized RPTCs (HK-2) and the epithelial-like pig kidney cell line (LLCPK-1) were maintained in a low glucose DMEM (01-050-1A; Biological Industries, Beit HaEmek, Israel) supplemented with 5% or 10% FCS, respectively. Cells were cultured at 37 °C in a humidified atmosphere of 5% CO_2_/95% air. To test the effect of CB_1_R activation/blockade, cells were challenged with one of the following synthetic cannabinoids: 5 µM AEA, 10 µM ACEA, 2.5 µM NE, 250 nM JZL-195, or 100 nM JD5037 for 24 h. At the end of the incubation period, the cells were harvested for further analyses.

### 2.8. Real-Time PCR

Total mRNA from in vivo studies (kidney homogenate) and in vitro studies (HK-2 or LLCPK-1 cells) was extracted using Bio-Tri RNA lysis buffer (Bio-Lab, Israel) followed by DNase I treatment (Thermo Fisher Scientific, Waltham, MA, USA). RNA purity was validated spectrophotometrically by nanodrop 2000 (Thermo Fischer Scientific, Waltham, MA, USA), and reverse transcribed using the iScript cDNA kit (Bio-Rad, Hercules, CA, USA). Real-time PCR analysis of 6 biological replicates in duplicates was performed using iTaq Universal SYBR Green Supermix (Bio-Rad, Hercules, CA, USA) and the CFX connect ST system (Bio-Rad, Hercules, CA, USA). The primers, used to assess the expression of various genes, were validated by blast sequencing, checked for specificity using the melting curve and Cq of the no template control (NTC) as well as optimized by defining the best annealing temperature, are listed in Table 1.

### 2.9. Western Blotting

Kidney and cell homogenates were prepared in a RIPA buffer (25 mM Tris-HCl, pH 7.6, 150 mM NaCl, 1% NP-40, 1% sodium deoxycholate, 0.1% SDS) using BulletBlender^®^ (Next Advanced, Inc., Troy, NY, USA) and zirconium oxide beads (Next Advanced, Inc., Troy, NY, USA). Protein concentrations were measured with the Pierce™ BCA Protein Assay kit (Thermo Fisher Scientific, Waltham, MA, USA). Cleared lysates were added with protein sample buffer, resolved by SDS-PAGE (4–15% acrylamide, 150V), and transferred to PVDF membranes using the Trans-Blot^®^ Turbo™ Transfer System (Bio-Rad, Hercules, CA, USA). The blots were then washed with 1× TBS-T (10 mM Tris-HCl, pH 7.6, 150 mM NaCl, 0.05% Tween20). Membranes were incubated for 1 h in 5% milk (in 1× TBS-T) to block unspecific binding, washed briefly, and incubated overnight at 4 °C with a primary antibody against EPO (ab224506; Abcam, Cambridge, UK) at a dilution recommended by the supplier (1:500). EPO antibody specificity was assessed by using its positive (kidney and liver) and negative (muscle and brain) controls as described in the literature [28]. The membrane was then washed with 1× TBS-T, and anti-mouse (ab98799; Abcam, Cambridge, UK) horseradish peroxidase (HRP)-conjugated secondary antibody was used for 1 h at room temperature, followed by chemiluminescence detection using Clarity™ Western ECL blotting substrate (Bio-Rad, Hercules, CA, USA). Densitometry was quantified using ImageLab software. Quantification was normalized to β-actin (ab49900; Abcam, Cambridge, UK), α-tubulin (DM1A; Cell Signaling Technology, Danvers, MA, USA), or valosin-containing protein (VCP) (ab204290; Abcam, Cambridge, UK), as reported in Table 2.

### 2.10. Immunohistochemistry

Pancreas tissues were fixed in buffered 4% formalin for 48 h and then embedded in paraffin. Sections were deparaffinized and hydrated. Heat-mediated antigen retrieval was performed with 10 mM citrate buffer pH 6.0 (Thermo Fisher Scientific, Waltham, MA, USA). Endogenous peroxide was inhibited by incubating with a freshly prepared 3% H_2_O_2_ solution in MeOH. Unspecific antigens were blockaded by incubating sections for 1 h with 2.5% horse serum (VE-S-2000, Vector Laboratories Inc., Burlingame, CA, USA). For assessing the cellular structure, pancreas sections were stained with guinea pig anti-insulin (A0564, Agilent Dako, Santa Clara, CA, USA; Table 3) antibody, followed by biotinylated secondary antibody and VECTASTAIN ABC reagent (VECTASTAIN ABC-Peroxidase kit, Vector laboratories). Color was developed after incubation with 3,3′-diaminobenzidine (DAB) substrate (ImmPACT DAB peroxidase (HRP) substrate, SK-4105, Vector Laboratories Inc., Burlingame, CA, USA), followed by hematoxylin counterstaining and mounting (Vecmount H-5000, Vector laboratories Inc., Burlingame, CA, USA). Stained sections were photographed using the LSM 700 imaging system (Zeiss, Oberkochen, Germany). Panoramic images were taken for the entire section using ZEN BLUE software (Zeiss, Oberkochen, Germany).

### 2.11. Statistical Analyses

Values are expressed as the mean ± SEM. An unpaired two-tailed *t*-test was used to determine differences between data sets comprised of two groups. Results in multiple groups were compared by ANOVA using Tukey’s multiple comparison test (GraphPad Prism v6 for Windows). Significance was set at *p* < 0.05.

## 3. Results

### 3.1. Nullification of CB_1_R in RPTCs Affects Bone Mass and Remodeling

To assess the possible involvement of RPTC-CB_1_R in the regulation of skeletal remodeling and mass, the skeletal phenotype of mice lacking CB_1_R in RPTCs was compared to that of their littermate wild-type (WT) controls. Trabecular bone parameters were analyzed at the distal femoral metaphysis, a well-established site for assessing both catabolic and anabolic stimuli [29]. Interestingly, RPTC-CB_1_^−/−^ mice were characterized by a high bone mass phenotype, manifested by a 17% increase in trabecular bone volume fraction (BV/TV), increased bone mineral density (BMD), and full femoral BV/TV (Figure 1A–D). The increase in trabecular bone volume density observed in the null animals was related primarily to significant changes in the trabecular thickness (Tb.Th; 9% increase), slightly increased trabecular number (Tb.N; Figure 1E,G), as well as an enhanced connectivity density (Conn.D), a parameter measuring the structural integrity of the trabecular network [30] (Figure 1F). Whereas a full bone length analysis revealed that the knockout mice had longer femora compared with their littermates (~0.4 mm increase; Figure 1H,J), no differences between the two mouse groups were found in any of the trabecular parameters in L3 vertebrae (Appendix A). Femoral mid-diaphysis analysis in RPTC-CB_1_^−/−^ mice revealed a reduction in the diameter of the medullar cavity (Med.Dia.) together with increased thickness of the cortical bone (Ct.Th.), without changes in the general diameter of the bone (diaphyseal diameter; Dia.Dia.) (Figure 1I,K–M).

To assess whether the high bone mass phenotype in RPTC-CB_1_^−/−^ mice is due to changes in bone formation or resorption, we next examined the presence and activity of osteoblasts and osteoclasts by using histomorphometric analyses. In fact, an enhanced mineral apposition rate (MAR; Figure 1N), representing increased osteoblast activity, without changes in the mineralizing surface (MS/BS; Figure 1P), resulted in a higher bone formation rate (BFR/BS; Figure 1O) in the RPTC-CB_1_ null mice. These measurements, however, were not manifested in changes in the circulating levels of osteocalcin, a protein secreted by active osteoblasts, since comparable levels were found between the two mouse strains (Figure 1Q). Using TRAP staining we counted osteoclasts and found a significantly higher ratio of osteoclasts per trabecular perimeter in the RPTC-CB_1_^−/−^ mice (N.Oc/BS; Figure 1R,S). However, their increased number did not result in a higher activity, as indicated by an equal amount of the circulating bone resorption marker CTX-1 [31] in both strains (Figure 1T). Altogether, these findings imply an important role for RPTC-CB_1_R in regulating bone mass accrual under normal physiological conditions.

### 3.2. CB_1_R Deletion in RPTCs Protected Mice from the Development of Diabetes-Induced Bone Loss

Considering the role kidney CB_1_R receptor has on bone homeostasis under healthy physiological conditions (Figure 1), we were interested in investigating whether its nullification in RPTCs has a protective effect on the skeleton in pathological conditions, specifically diabetes. To that end, we used a T1D mouse model, induced by exposing the mice to STZ (50 mg/kg IP for 5 days). The efficiency of the diabetic model was assessed by verifying that all diabetic mice were hyperglycemic (glucose levels > 250 mg/dL), hypoinsulinemic, as well as by assessing Langerhans islet morphology (Appendix A). µCT analysis of the distal femoral metaphysis revealed significant reductions in bone mass volume, mineral content, and full bone density in the diabetic WT animals compared with their non-diabetic controls. Interestingly, no reductions in any of these skeletal parameters were observed in the diabetic RPTC-CB_1_^−/−^ mice when compared either to their non-diabetic controls or their WT littermate controls (Figure 2A–D). The diabetic-induced trabecular bone loss was attributed to significant reductions in Tb.Th (~−36%; Figure 2E) and Tb.N (~−16%; Figure 2H). There were insignificant trends toward increased trabecular spacing (Tb.Sp) and reduced Conn.D (Figure 2F,G), parameters that remained normal in the diabetic knockout mice. The protective effect of CB_1_R nullification during hyperglycemia was further extended to the cortical bone, demonstrating a reduction in femoral mid-diaphyseal Ct.Th in diabetic WT animals that was absent in the diabetic RPTC-CB_1_^−/−^ mice (Figure 2I–L). Moreover, µCT analysis of L3 vertebrae revealed a similar protection in bone mass and density in mice lacking CB_1_R in RPTCs (Appendix A).

Using a 3-point bending test of the femurs, we next found that hyperglycemia resulted in reduced femoral stiffness, fracture force, and maximal force in diabetic WT mice compared with their non-diabetic controls. These findings were additionally supported by the reduction in the moment of inertia (MOI) values in the diabetic WT group by utilizing µCT (Figure 2P). No changes in any of these parameters were found in the diabetic RPTC-CB_1_^−/−^ animals (Figure 2M–P), suggesting that knocking out CB_1_R in RPTCs protects the animals from the deleterious effects of hyperglycemia on both trabecular and cortical bone, and preserves their biomechanical properties and resistance to diabetic fractures.

Histomorphometrically, we found that hyperglycemia significantly reduced osteoblast activity (MAR) and number (MS/BS), resulting in a reduction of BFR as well as reduced circulating osteocalcin levels in WT animals. No such effects, however, were found in the diabetic RPTC-CB_1_^−/−^ mice when compared either to their non-diabetic controls or to their WT littermate controls (Figure 3A–D). In parallel, enhanced osteoclastogenesis (Figure 3E–G), coupled with elevated circulating levels of non-specific bone turnover biomarkers, such as CA2 (Figure 3H), PHOS (Figure 3I), and ALP2S (Figure 3J), were found in the diabetic WT animals; normal values of these markers were found in the null mice. Interestingly, urinary PHOS secretion was increased under diabetic conditions regardless of genotype (Appendix A). Additionally, no changes in the PHOS transporters’ density were observed between the diabetic WT and RPTC-CB_1_^−/−^ mice (Appendix A), indicating that the changes in circulating PHOS levels are kidney independent. Taken together, these findings suggest a role for RPTC-CB_1_R in remotely regulating osteoblast and osteoclast function.

### 3.3. Pharmacological CB_1_R Blockade Prevents T1D-Induced Bone Loss

Complementary to the results described above in RPTC-CB_1_^−/−^ mice, chronic treatment with SLV-319 (3 mg/kg, orally for 16 weeks), a globally acting CB_1_R blocker, attenuated the bone loss in an STZ-induced T1D model (Figure 4) without affecting either body weight or systemic glucose homeostasis (Appendix A). Specifically, analysis of the distal femoral metaphysis BV/TV in vehicle-treated diabetic mice was found to be markedly reduced (~−26%) in comparison with the vehicle-treated naïve non-diabetic animals (Figure 4A,B). This reduction was completely attenuated in the SLV-319-treated animals (Figure 4A,B), primarily due to the significantly increased Tb.N and Conn.D as well as to decreased Tb.Sp (Figure 4E–H). Interestingly, these effects did not apply to full bone parameters such as full bone BV/TV, BMD, or Ct.Th, which remained unchanged in the SLV-319-treated animals (Figure 4C,D,I). The positive effects of SLV-319 on trabecular bone values were also recapitulated in L3 vertebrae with enhanced BV/TV values due to increased Tb.Th and Tb.N (Appendix A).

Histomorphometric assessment of bone turnover parameters revealed that the reversal of bone loss in SLV-319-treated diabetic WT mice resulted from both decreased osteoclast number and increased bone formation. Calcein labeling analysis showed that the reduction in all of the bone formation parameters, including MAR, BFR, and mineralizing surface, associated with diabetes, were abated in the SLV-319-treated group (Figure 4J–M). These findings were further supported by the elevation in osteocalcin levels as a result of SLV-319 treatment (Figure 4N). Furthermore, SLV-319 mitigated the increase in bone resorption, manifested in a decrease in the number of TRAP+ osteoclasts normalized to the bone surface (Figure 4O–Q). However, no changes in the CTX-1 levels were noted in the SLV-319-treated group (Figure 4R). Interestingly, SLV-319 did not reduce the hyperglycemia-induced increase in the non-specific markers of bone turnover (Figure 4S–U). Taken together, our data suggest that CB_1_R blockade may have the therapeutic potential to prevent hyperglycemia-induced bone loss.

### 3.4. CB_1_R in RPTCs Regulates EPO Levels In Vivo

Encouraged by the findings that genetic deletion of CB_1_R in RPTCs and its pharmacological blockade by SLV-319 protected mice from the deleterious effects of hyperglycemia on the skeleton, we next tried to suggest a molecular effector linking kidney CB_1_R with bone remodeling and mass. Since the proximal tubule [32] is the ideal location for EPO production by a subset of peritubular fibroblasts [33,34], which is known to negatively affect skeletal remodeling (reviewed in [35]), we first analyzed its protein levels in kidney samples collected from animals whose skeletal phenotype was assessed (Figure 5). Interestingly, we found a significant elevation in renal EPO levels in the diabetic WT mice compared with their non-diabetic controls. No similar elevation in its levels was found in the diabetic RPTC-CB_1_^−/−^ mice (Figure 5A). To determine whether CB_1_R directly regulates kidney EPO levels, we measured the renal protein expression levels of EPO in mice treated acutely (6 h) with the CB_1_R activator, ACEA (10 mg/kg, IP). As shown in Figure 5B, the EPO protein levels were significantly elevated in the kidney following stimulation of the CB_1_R, suggesting a novel mechanism by which the EPO levels are regulated.

### 3.5. CB_1_R in RPTCs Modulates EPO Levels In Vitro

Next, we examined the effect of CB_1_R activation or blockade on EPO levels in vitro. Interestingly, we found that activation of CB_1_R in two kidney cell lines, HK-2 and LLCPK-1, either by using direct agonists of CB_1_R (NE, AEA) or by indirectly elevating eCB levels (using JZL-195), resulted in an elevation in the mRNA (Figure 6A,C,E,G) and protein (Figure 6B,D,F,H) levels of EPO. However, these effects were inhibited by pretreating the cells with JD5037, a potent CB_1_R antagonist.

Since CB_1_R is G protein-coupled receptor (GPCR), coupled to the Gi/o subclass of G-proteins, we next evaluated the signal transduction connecting CB_1_R activation to EPO. To that end, we pretreated the cells with PTX, which inhibits the Gi-signaling pathway, prior to treating them with NE, and found that NE lost its ability to upregulate EPO protein expression (Figure 6F), suggesting a Gi/o-signal transduction mechanism by which CB_1_R regulates EPO levels.

## 4. Discussion

Bone development, remodeling, and metabolism are well known to be regulated by various local, central, and systemic factors, which are either generated or their activity is regulated by the kidney. Whereas normal kidney function is important for bone health, conditions that deteriorate renal function, such as DKD may ultimately lead to impairment in bone homeostasis and remodeling. Therefore, elucidating novel targets in the kidney that may affect bone regulators is therapeutically relevant in patients with DKD. Here, we suggested a novel kidney-to-bone axis in which CB_1_R in RPTCs regulates bone homeostasis via affecting EPO levels, demonstrating that genetic nullification or pharmacological blockade of CB_1_R has a protective effect on the skeleton of diabetic mice by downregulating the availability of EPO.

CB_1_R has been extensively explored in modulating both bone and kidney homeostasis. In bone, CB_1_R is expressed in various bone cell types as well as in macrophages, fat cells, and skeletal sympathetic nerve terminals located in small proximity to the bone cells (reviewed in [36]). Its effects on bone remodeling and mass were found to be age and strain dependent [14,37,38], suggesting that CB_1_R plays a multifaceted role in directly modulating bone remodeling and mass. Our current findings add to this complexity by demonstrating that specific nullification of RPTC-CB_1_R in male mice on a C57Bl/6J background results in a high peak bone mass phenotype and protected the animals against diabetes-induced osteoporosis. In the kidney, CB_1_R also plays a pivotal role in regulating renal hemodynamics, inflammation, and fibrogenesis in pathologic states such as DKD and obesity-induced renal dysfunction. Specifically, we and others have shown that pharmacological blockade or genetic deletion of CB_1_R in RPTCs protects the kidney from the deleterious effects of hyperglycemia (in T1D and T2D) and/or fatty acid flux (obesity), and ameliorates diabetes- and obesity-induced albuminuria, fibrosis, and renal inflammation [19,20,21,22,39,40]. These effects were mediated via CB_1_R located either in podocytes and/or RPTCs, since their specific nullification from these cells completely prevented the development of hyperglycemia- or lipotoxicity-induced renal dysfunction.

Since the kidney is a major source of eCBs, whose levels are elevated during diabetes and obesity [22,41], and the fact that CB_1_R, highly expressed in many cells within the kidney, is also upregulated in these conditions [21,22,42,43,44,45,46,47], the possible involvement of the renal eCB/CB_1_R system in regulating skeletal remodeling and mass in health and disease led us to test the hypothesis that a negative axis, involving CB_1_R in RPTCs, exists between kidney and bone. Indeed, nullification of CB_1_R from the RPTCs under normoglycemic conditions led to a high peak bone mass phenotype in mice, which can be attributed to increased BFR rather than reduced osteoclastogenesis, since enhanced osteoblast activity was measured in vivo in RPTC-CB_1_^−/−^ mice. Osteocalcin, secreted almost exclusively by osteoblasts [48,49], was not significantly different between the two mouse strains. This may suggest that there was no difference in osteoblast number, further supporting the findings of unchanged MS/BS, a surrogate of osteoblast number. Surprisingly, we found a high number of osteoclasts in the RPTC-CB_1_^−/−^ mice, despite the fact that they had a high bone mass phenotype. The unchanged levels in the bone resorption marker CTX-1 [49] may suggest that although the osteoclast number was high, their activity rate was most likely inhibited. Additionally, since CTX-1 levels were not normalized to bone surface they may not truly reflect the bone resorption rate.

Next, we found that deleting CB_1_R in RPTCs significantly hindered T1D-induced bone loss in comparison with their diabetic WT controls. The reduction in bone mass observed in the diabetic WT animals, supported by previous findings in diabetic patients and mouse models [50,51,52,53,54], was mainly attributed to a significant reduction in the trabecular and cortical bone parameters, which were further correlated with reduced resistance to fracture. Our findings, which indicate a complete protection from the development of T1D-induced bone loss in animals lacking CB_1_R in RPTCs, suggest a novel therapeutic target. Indeed, similar findings were found in diabetic mice treated with SLV-319, which ameliorated hyperglycemia-induced bone loss without affecting systemic glucose homeostasis. These skeletal improvements are most likely related to the preserved renal function in these animals, as we reported previously [20], and further support a kidney-to-bone axis. In line with our findings, the CB_1_R blockade was found to ameliorate bone morphological changes induced by chronic intermittent hypoxia in rats [55], as well as to attenuate a glucocorticoid-induced reduction in bone mineral density via improving osteoblast function and reducing bone marrow adiposity [56]. Interestingly, these positive effects of CB_1_R blockade were age dependent [57]. Impaired osteoblast activity and number are central to bone loss and fragility seen in T1D. Since bone formation is known to be regulated by various factors including mesenchymal stem cell differentiation to adipocytes over osteoblasts as well as by affecting osteoblast fate, others demonstrated that hyperglycemia results in elevation in bone marrow adiposity and osteoblast death [52,58,59,60]. Although these parameters were not assessed here, our findings of reduced MAR and MS/BS, which were manifested by deceased BFR in diabetic WT mice, support the evidence that maintaining osteoblast number and function is crucial for preventing T1D-induced bone loss. In fact, the abovementioned molecular abnormalities, together with the reduction in the circulating osteocalcin levels, were almost fully attenuated in diabetic mice lacking CB_1_R in RPTCs or that were treated with SLV-319.

Unlike the effect of hyperglycemia on osteoblasts, the current evidence that examined the role of T1D in affecting osteoclast number and function remains controversial. Our findings indicate that pharmacological blockade or genetic deletion of CB_1_R specifically in RPTCs significantly attenuated the increased osteoclast number without changing the CTX-1 levels. Comparable findings, demonstrating an insignificant effect of hyperglycemia on the serum levels of CTX-1 with an increased number of osteoclasts, were also reported by others [61,62]; however, a study by Guo and colleagues demonstrated that STZ increases both the osteoclast number and the CTX-1 levels [63]. In support of the osteoporotic phenotype in the diabetic WT animals, which was ameliorated by CB_1_R deletion or blockade, one should note the elevated circulating levels of PHOS and ALPL, which are known to be correlated with an increased bone turnover phenotype [64,65,66,67]. Interestingly, decreased bone quality in hyperglycemic conditions was even associated with no change or even a reduction in bone resorption [51,60,68], further suggesting that osteoblasts, rather than osteoclasts, play a leading role in determining the bone phenotype in diabetic patients and animals.

Among the various kidney-related bone modulators, recent evidence supports the critical role of EPO in regulating bone homeostasis either directly or systemically (reviewed in [35]). Since EPO is generated by interstitial fibroblast in close proximity to the RTPCs [33,34] as well as by tubular cells [32], and the fact that the eCB system affects RPTC’s function via modulating the activity of CB_1_R [19,20,22], it was only natural for us to further investigate the involvement of RPTC-CB_1_R in influencing bone impairment during diabetes via affecting EPO levels. In accordance with renal hypoxia being a common feature of DKD [69,70], and the fact that HIF-1α promotes renal EPO synthesis in the RPTCs in diabetic humans and animals [71,72,73], we found a significant upregulation in renal EPO expression levels in diabetic WT animals that was completely absent in the RPTC-CB_1_^−/−^ mice. An opposite effect, demonstrating an upregulation of renal EPO levels, was found in non-diabetic WT animals treated with the specific CB_1_R agonist ACEA. In fact, stimulating CB_1_R has been previously shown to induce hypoxia in the kidney and other tissues, whereas CB_1_R deletion/blockade ameliorated it [39,55,74,75]. These changes have not been linked before to EPO homeostasis; therefore, our findings suggest that CB_1_R is a novel regulator of EPO synthesis. To further demonstrate a direct regulation of EPO by CB_1_R in RPTCs, we utilized two cell lines (from different sources: Human and pig), and showed that indeed the gene and protein levels of EPO are elevated following the activation of CB_1_R. Furthermore, indirectly activating CB_1_R, by using JZL-195, which inhibits FAAH and MAGL, the enzymes responsible for AEA and 2-AG degradation, respectively, resulted in a similar upregulation in EPO levels, an effect that was blocked by the CB_1_R antagonist JD5037. These effects were Gi-dependent, since NE did not affect the EPO levels in the presence of PTX. A previous study reported that AEA increased the EPO-induced stimulation of hematopoietic cell proliferation [76]; however, this effect was attributed to the role of AEA in activating CB_2_R. Therefore, the current study is the first to describe a direct relationship between CB_1_R and EPO.

Based on our current findings, it is reasonable to assume that activating CB_1_R increases the production and secretion of EPO to the serum, which in turn, may lead to bone loss. This hypothesis, reinforced by our findings, is based on the growing evidence that EPO negatively affects skeletal remodeling, and that its levels were found to be correlated with a low trabecular bone mass phenotype [35,77,78,79]. EPO, a heavily glycosylated protein that acts by activating the erythropoietin receptor (EPO-R), is primarily known for its role in hematopoiesis [80]. Following the discovery of EPO-R outside the hematopoietic tissues, EPO has been widely studied for its possible non-hematopoietic effects, such as skeletal homeostasis. Interestingly, its role in regulating bone maintenance and regeneration remains controversial. Whereas our findings support the existing evidence of EPO in the negative regulation of bone remodeling and mass, others have shown that EPO directly activates mesenchymal stem cells to form osteoblasts in vitro; this in turn, increases bone formation in vivo [81]. Additionally, studies have repeatedly shown that the viability and proliferation of osteoblasts are enhanced by EPO [82], which is also known to improve bone healing [83]. However, there is still a debate whether the accelerated bone healing induced by EPO is mediated via its ability to induce angiogenesis, a crucial step in fracture healing, or by directly influencing osteoblasts [35,84]. Our findings add to the existing controversy regarding the role of EPO in regulating bone mass; however, they are in line with previous studies showing that EPO directly stimulates osteoclast precursors, induces bone loss, and inhibits murine osteoblast differentiation [77]. Nevertheless, further studies will need to shed more light on how EPO affects the activity of osteoblasts and osteoclasts, and more specifically during diabetes. 

The current study suggests a role for RPTC-CB_1_R in regulating bone remodeling and mass via EPO. However, it does not rule out the plausible roles of additional factors, such as 1,25(OH)_2_D_3_, Klotho, BMP-7, PTH, osteoprotegerin, sclerostin, and FGF-23, which influence bone mass and remodeling and could also play an important role in the skeletal phenotype described here. In fact, several studies reveal their contribution to diabetes-induced osteoporosis [85,86,87,88,89,90,91], but further studies are warranted to assess whether these factors are also controlled by the eCB/CB_1_R system.

## 5. Conclusions

This work presents a possibly novel kidney-to-bone axis modulated via RPTC-CB_1_R. Nullification of CB_1_R from the RPTCs preserved bone mass under hyperglycemic conditions. This was achieved via affecting both osteoclastogenesis and bone formation. Although the exact mechanism underlying this phenomenon remains unknown, we suggest that EPO, generated by the kidney, is most likely involved, since its role in bone metabolism has been repeatedly demonstrated, and its levels are elevated during diabetes and are explicitly regulated by RPTC-CB_1_R. From a pharmacological point of view, the antagonism of CB_1_R in RPTCs has a therapeutic potential in treating diabetes-induced osteoporosis, and in improving the quality of life of patients suffering from diabetes.

## Figures and Tables

**Figure 1 cells-10-00414-f001:**
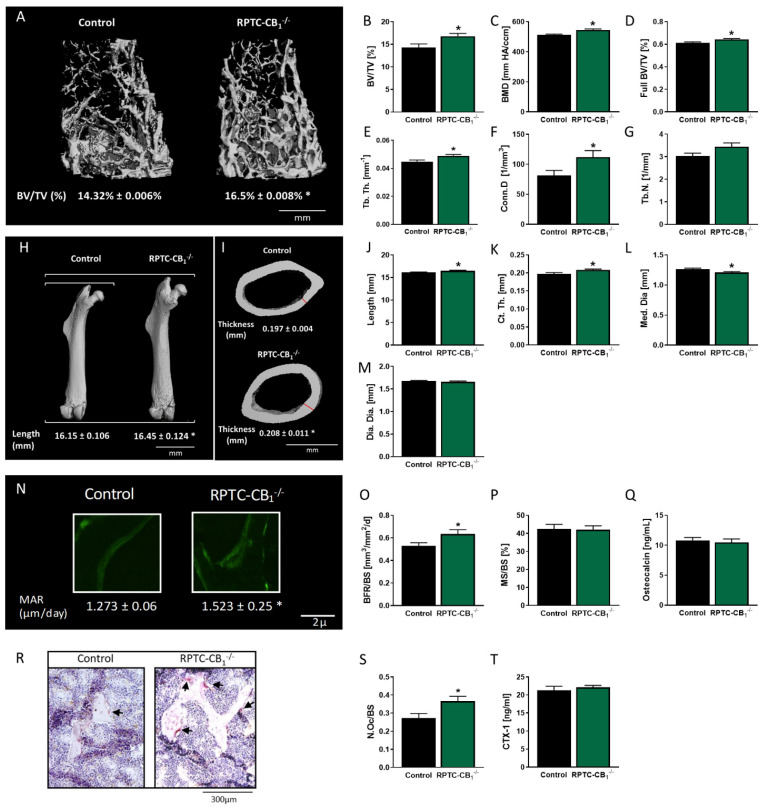
Main endocannabinoid receptor (CB_1_R) in renal proximal tubule cells (RPTCs) modulates bone mass and function under normoglycemic conditions. 3D images of the distal femoral metaphysis of RPTC-CB_1_^−/−^ mice and their littermate wild-type (WT) controls with median bone volume density (BV/TV) values (**A**,**B**), bone mineral density (BMD; **C**), full bone BV/TV (**D**), trabecular thickness (Tb.Th; **E**), connectivity density (Conn.D; **F**), and trabecular number (Tb.N; **G**). 3D images of the femora of RPTC-CB_1_^−/−^ mice and their littermate WT controls with median values of length for each group (Length; **H,J**). 3D images of the femoral mid-diaphysis of mice with median values of cortical thickness (Ct.Th) for each group (**I,K**), medullary diameter (Med.Dia; **L**), and diaphyseal diameter (Dia.Dia; **M**). Histomorphometric analysis of the bone formation parameters. Representative images of calcein-labeled mineralized fronts with mineral apposition rate (MAR) values (**N**), the bone formation rate (BFR/BS; **O**), mineralizing surface (MS/BS; **P**), and serum osteocalcin levels (**Q**). Images of TRAP+ osteoclasts (**R**), the number of osteoclasts per trabecular surface area (N.Oc/BS; **S**), and the serum levels of CTX-1 (**T**). Data represent the mean ± SEM from 10–14 animals per group, * *p* < 0.05 relative to the corresponding control group.

**Figure 2 cells-10-00414-f002:**
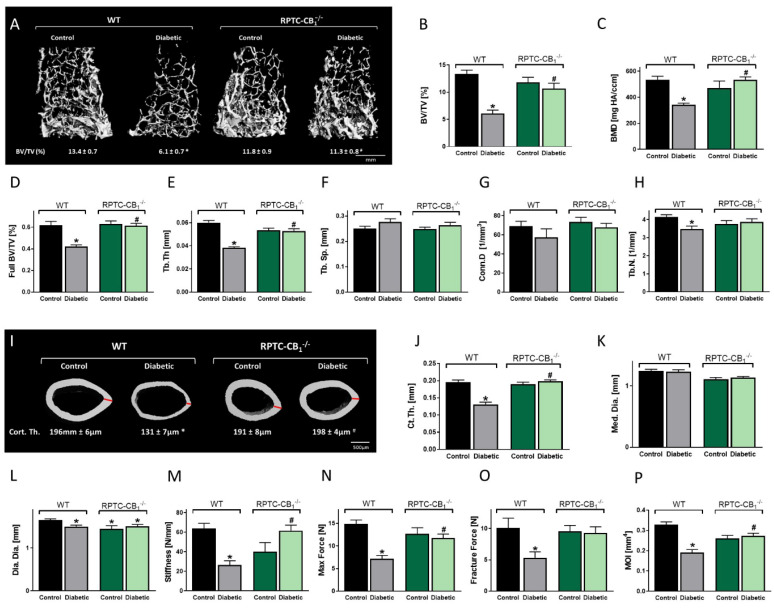
Nullification of CB_1_R in RPTCs protects against diabetes-induced bone loss. 3D images of the distal femoral metaphysis of RPTC-CB_1_^−/−^ mice and their littermate WT controls with median bone volume fraction (BV/TV) values (**A**,**B**), bone mineral density (BMD; **C**), full bone BV/TV (**D**), trabecular thickness (Tb.Th; **E**), trabecular spacing (Tb.Sp; **F**), connectivity density (Conn.D; **G**), and trabecular number (Tb.N; **H**). 3D images of the femoral mid-diaphysis of mice with median values of cortical thickness (Ct.Th) for each group (**I**), cortical thickness (Ct. Th; **J**), medullary diameter (Med.Dia; **K**), diaphyseal diameter (Dia.Dia; **L**), stiffness (**M**), maximal force (**N**), fracture force (**O**), and moment of inertia (MOI; **P**). Data represent the mean ± SEM from 8–13 animals per group, * *p* < 0.05 vs. non-diabetic WT control mice, ^#^
*p* < 0.05 vs. diabetic WT mice.

**Figure 3 cells-10-00414-f003:**
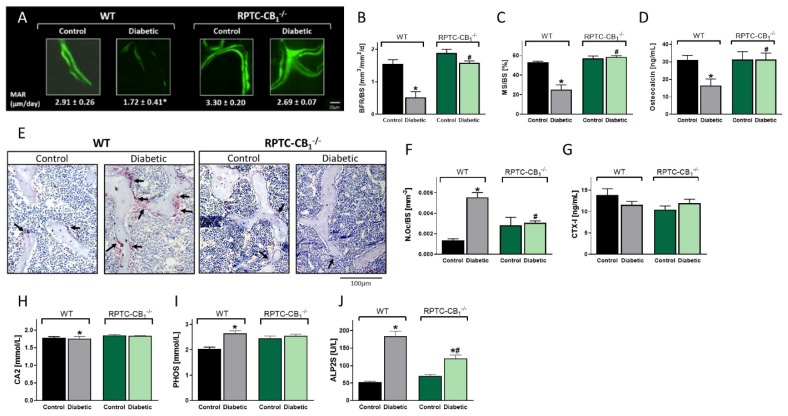
Nullification of CB_1_R in RPTCs prevents diabetes-induced inhibition of bone formation and upregulation in osteoclastogenesis. Representative images of calcein-labeled mineralized fronts with mineral apposition rate (MAR) values (**A**). Bone formation rate (BFR/BS; **B**), mineralizing surface (MS/BS; **C**), and osteocalcin serum levels (**D**). Images of TRAP+ osteoclasts (**E**), the number of osteoclasts per trabecular surface area (N.Oc/BS; **F**), and the serum levels of CTX-1 (**G**). The levels of the circulating biochemical markers, calcium (CA2; **H**), phosphate (PHOS; **I**), and alkaline phosphatase (ALP2S; **J**). Data represent the mean ± SEM from 8–13 animals per group, * *p* < 0.05 vs. non-diabetic WT control mice, ^#^
*p* < 0.05 vs. diabetic WT mice.

**Figure 4 cells-10-00414-f004:**
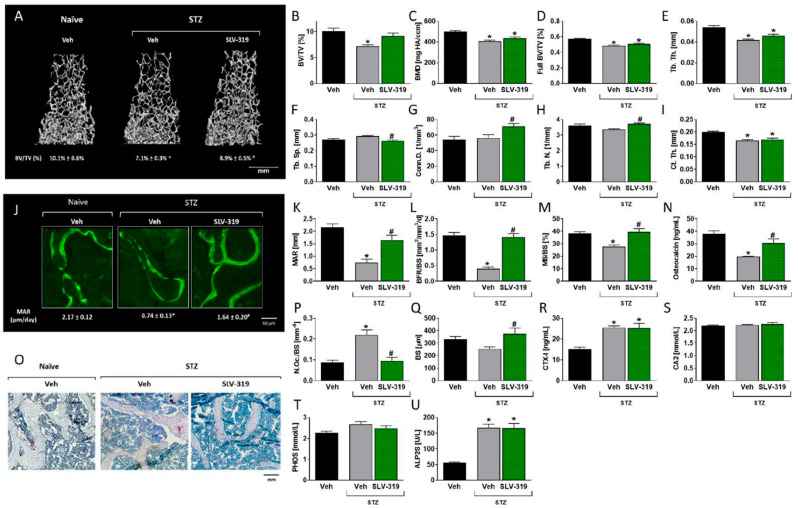
Chronic CB_1_R blockade prevents T1D-induced bone loss. 3D representative images of the distal femoral metaphysis of mice with median bone volume density (BV/TV) values (**A**,**B**). Bone mineral density (BMD; **C**), full bone BV/TV (**D**), trabecular thickness (Tb.Th; **E**), trabecular spacing (Tb.Sp.; **F**), connectivity density (Conn.D; **G**), trabecular number (Tb.N; **H**), and cortical thickness (Ct.Th; **I**). Histomorphometric and biochemical analysis of bone remodeling parameters reveals a rescue of diabetes-induced bone loss by SLV-319. Representative images of calcein-labeled mineralized fronts (**J**), the mineral apposition rate (MAR; **K**), the bone formation rate (BFR/BS; **L**), the mineralizing surface (**M**), and the serum osteocalcin levels (**N**). Images of TRAP+ osteoclasts (**O**) show the number of osteoclasts per trabecular surface area (N.Oc/BS; **P**), the bone surface (BS; **Q**), and the serum levels of CTX-1 (**R**), as well as the levels of biochemical markers, calcium (CA2; **S**), phosphate (PHOS; **T**), and alkaline phosphatase (ALP2S; **U**). Data represent the mean ± SEM from 8–10 mice per group, ** p* < 0.05 relative to the corresponding control group treated with Veh, *^#^ p* < 0.05 relative to the corresponding streptozotocin (STZ) group treated with Veh.

**Figure 5 cells-10-00414-f005:**
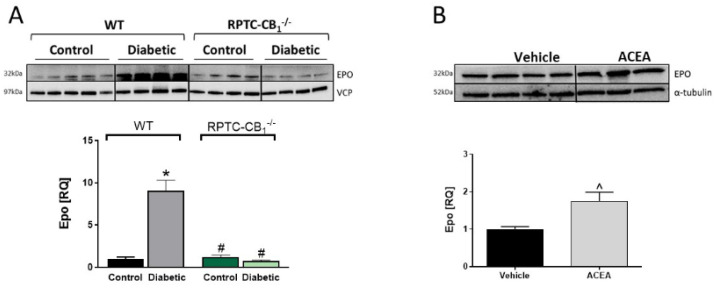
Renal EPO levels are elevated during diabetes or by activation of CB_1_R. Representative Western blots and quantification of the protein levels of EPO in kidney samples collected from normal and diabetic WT and RPTC-CB_1_^−/−^ mice (**A**), representative Western blots and quantification of renal protein levels of EPO in mice treated acutely with arachidonyl-2′-chloroethylamide (ACEA) (10 mg/kg, IP; **B**). Data represent the mean ± SEM from 8–13 animals per group, * *p* < 0.05 vs. non-diabetic control WT mice, ^#^
*p* < 0.05 vs. diabetic WT mice, ^ *p* < 0.05 vs. vehicle-treated mice.

**Figure 6 cells-10-00414-f006:**
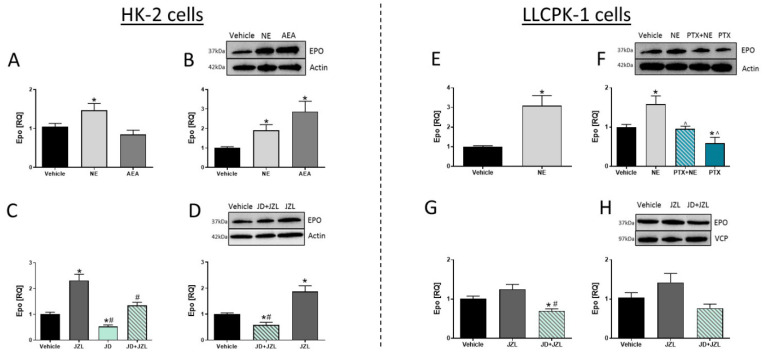
CB_1_R modulates EPO levels in vitro. mRNA expression levels of EPO in HK-2 cells (**A**,**C**). Representative Western blots and quantification of EPO in HK-2 cells (**B**,**D**). mRNA expression levels of EPO in LLCPK-1 (**E**,**G**). Representative Western blots and quantification of EPO in LLCPK-1 cells (**F**,**H**). Data represent the mean ± SEM from 2–4 independent experiments conducted in 6 replicates in each treatment group, * *p* < 0.05 vs. vehicle, ^#^
*p* < 0.05 vs. JZL-195, ^ *p* < 0.05 vs. noladin ether (NE).

**Table 1 cells-10-00414-t001:** Primers used for qPCR analysis.

Gene	Forward Primer (5′–3′)	Reverse Primer (5′–3′)
hRPLP0 *homo sapience*	AATCCCATCACCATCTTCCA	TGGACTCCACGACGTACTCA
hEPO *homo sapience*	CATGTGGATAAAGCCGTCAGT	AAGTGTCAGCAGTGATTGTTCG
pGAPDH *sus scrofa*	CACGACCATGGAGAAGGC	GAAGCAGGGATGATGTTCTGG
pEPO *sus scrofa*	GCAAGTCGAAACCTGAGCTG	ACTTGTCCCGGCCAAACTT
mUbc *mus musculus*	GCCCAGTGTTACCACCAAGA	CCCATCACACCCAAGAACA
mSlc34a1 *mus musculus*	GGCTCCAACATTGGCACTAC	ACAGTAGGATGCCCGAGATG
mSlc34a2 *mus musculus*	GCTTGACTTAGGGCAGGTGTGG	AGGGGCTCAGTTTGGCATCTC
mSlc34a3 *mus musculus*	TACCCCCTCTTCTTGGGTTC	CAGTCTCAAGACAGGCACCA
mSlc20a2 *mus musculus*	GTGCCGGCCCTGCTTAC	CAATGCCTCTGCTTTCGTTCT
mSlc17a1 *mus musculus*	ACCCGTATATGAGCAGCAGTGAGA	AAATGTCGGCGTGTATGTAACCAG

**Table 2 cells-10-00414-t002:** List of antibodies used for Western blotting.

Protein	Reference	Supplier
EPO	ab224506	Abcam
β-actin	ab49900	Abcam
α-tubulin	DM1A	Cell signaling technology
VCP	ab204290	Abcam

**Table 3 cells-10-00414-t003:** List of antibodies used for immunohistochemistry.

Protein	Reference	Supplier
Insulin	A0564	Agilent Dako

## Data Availability

The data presented in this study are available on request from the corresponding author.

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
