# Peer review of "Renal Proximal Tubule Cell Cannabinoid-1 Receptor Regulates Bone Remodeling and Mass via a Kidney-to-Bone Axis"

_cells, 2021, doi:10.3390/cells10020414_

Round 1
Reviewer 1 Report
Overall, this is a nice study and a well written paper. However, there are several concerns that should be addressed. Generally, there is a lack of methodological detail and the Discussion is extensively long. Specific comments are given below.
The Introduction is very well written.
Section 2.2. Please explain what SLV319 is and what it does. Calcein is used for assessing mineralization, not formation per se. That should be clarified. The calcein schedule seems quit long if I understand it correctly. It seems injections where given 120 days apart, is that right? References should be used to justify the choice of the labeling schedule.
Section 2.3. Please describe what ACEA, NE, and AEA are and what they are used in the experiment for.
Section 2.4. Please describe if you measured total and/or ionized calcium. Many serum markers of bone remodeling (eg, osteocalcin and ctx) accumulate with kidney disease and are therefore unreliable. Please provide rationale for the choice of remodeling markers instead of markers that do not accumulate with KD like BSALP and TRAPC.
Section 2.5. Please provide μCT/histomorphometric methodology and outcome variables here, at least briefly with all essential information, like the scanning parameters, voi definitions, and define uncommon outcome variables like “full bone BV/TV” and “Dia.Dia”. The cross-sectional moments of inertia for the femurs should be reported as those are more indicative of bone strength.
Section 2.6. Likewise please provide the details, including which bone was tested, how it was stored prior to testing, span length/femur length (presuming it was the femur), use of beam theory, strain rate, outcome variables, etc…
Section 2.7. Please indicate why human and pig cells instead of mouse cells were used. Why not do the analyses in the experimental mice? Please provide rationale in the paper. Please indicate if the analyses in 2.8, 2.9, and 2.10 are just for the in vitro work and/or the in vivo experiments.
Section 2.8. Please provide brief rationale for choice of genes to assess.
Section 2.9. Please provide brief rationale for choice of proteins to assess.
Section 2.11. Please describe the experimental designs (eg, number of groups, samples sizes, etc) for the in vivo and in vitro experiments so it’s implicit which sets of results were analyzed by t-test vs Anova, or explicitly state which data sets got which analyses. Bonferroni’s is only used to compare selected pairs of means. Please describe which selected pairs you compared and why. Otherwise, maybe Tukey’s should be used.
Section 3.1 begins with too many “Introduction” type statements. It would be easier to follow if only results from this study are concisely and clearly presented in the Results section.
Please give some quantitative differences in the text for some of the key outcome variables so that the reader can assess if statistically significant differences are likely to have physiological significance. For example, what is the difference in mm between the groups in Fig 1 K?
BV/TV has been shown to be correlated with TbTh (PMID: 10404017). Its hard to imagine how BV/TV increases when TbTh abd TbN do not. Hopefully that is addressed in the Discussion.
N.Ob/BS should be reported to complete the story with Fig 1 O-T.
Section 3.2: There is unnecessary repetition of Intro/Methods. The entire first paragraph is unnecessary. It will be easier to follow if only results are presented clearly and concisely in the Results section.
This statement makes sense, but further adds to the confusion of the findings in 3.1: “ The diabetic-induced trabecular bone loss was attributed to significant reductions in Tb.Th (Figure 2E) and Tb.N (Figure 2H).”
How was MOI calculated? It seems there may be a mistake because the units are incorrect.
Are the WT controls and RPTC-CB-/- controls in Figs 2 and 3 the same comparisons as in Fig 1? If so, it would be much easier to follow by omitting Figure 1 and corresponding text, and make those comparisons using figs 2 and 3.
N.Ob/BS should be reported to see how it compares with osteocalcin.
Apparently, CTX isn’t representative of your histological findings at this particular skeletal site. That should be addressed in the Discussion.
Section 3.4: What is the evidence for hyperglycemia? “blockade by SLV-319 protected mice from the deleterious effects of hyperglycemia on the skeleton”
Section 3.6 should be renamed 3.5. It also should not include methods. The methods should be described in the Methods section instead.
Discussion:
The discussion is extensively long and should be made considerably more concise. Staying focused on the work in this study (eg, CB1 and EPO) would make it easier to follow. For example, most of paragraph 1 could be omitted, the CB1/EPO parts could be merged concisely with the second paragraph. Similarly, paragraphs 3 and 4 could be merged into a single, much shorter, concise paragraph.
An acknowledgement and discussion of the limitations with the study should be provided near the beginning of the Discussion. For example, the limitations with serum markers and the use of caution in interpretation should be discussed.
A figure describing a proposed possible mechanism would be helpful: Starting with the roles of FAAH and MGL, what causes the changes in ECs, how ECs act on RPTCs, how that affects EPO….changes in bone remodeling and structure.
Reviewer 2 Report
The authors have found interesting effects on bone that appear to mediated by proximal tubule endocannabinoid receptor activation. In general, the studies appear to be well performed and the results appear convincing. However, I have some further suggestions.
The proximal tubule is known to regulate phosphate reabsorption through FGF23 and PTH mediated actions. It would be important to provide information about the role of endocannabinoids on FGF23 and PTH levels, whether proxmal tubule phosphate transporter density is altered and whether there is altered phosphaturia/proximal tubule phosphate reabsorption with CB1 deletion.
It is not actually correct to state that the proximal tubule is a source of EPO. EPO in the kidney is produced by a subpopulation of interstitial fibroblasts that are in close proximity to the proximal tubule, and studies have shown that alterations in proximal tubule function can affect EPO production by these cells.
Reviewer 3 Report
In this manuscript from Dr Joseph Tam's group, the authors report that genetic deletion or pharmacological inactivation of CB1 on renal proximal tubule cells preserves bone mass under normoglycemic as well as hyperglycemic conditions. This effect is presumably through modulation of EPO, which is a well-known regulator of bone health. Overall, this is an interesting paper that builds upon previous reports that is suggestive of a protective role of CB1 antagonism in preserving bone density. I advocate publication pending modifications below:
- Materials and methods -- (1) Move materials to the top of the section (2) List the source of transgenic animals and how they were genotyped, ideally provide primer sequences used for this work (3) Provide BW/glucose levels of animals as supplementary data for key studies or at a minimum please comment on what these were (4) RT-PCR -- were the primers adequately tested for linearity? Were MIQE guidelines followed? Ideally, should provide linearity testing data for at least EPO since this is key (5) Western blotting -- spell out VCP
- Results -- A fairly large distribution of animals were used (8-13 and 6-10). Were unequal group sizes used? Please clarify. (3.6) Please consider providing RT-PCR or protein data to demonstrate that CB1 is expressed in these cell lines. (edit) NE lost its ability to upregulate the protein expression of EPO to NE lost its ability to upregulate EPO protein expression.
- Discussion -- could be a little concise
Round 2
Reviewer 1 Report
Thanks for addressing the previous concerns
Reviewer 2 Report
The authors have provided additional studies that improve the manuscript. I have no further concerns.